# Antifungal Efficacy of Luliconazole-Loaded Nanostructured Lipid-Carrier Gel in an Animal Model of Dermatophytosis

**DOI:** 10.3390/jof11040324

**Published:** 2025-04-19

**Authors:** Robab Ebrahimi Barogh, Seyyed Mobin Rahimnia, Mohsen Nosratabadi, Abolfazl Maleki, Fatemeh Khosravi Ebrahimi, Zahra Yahyazade, Iman Haghani, Pedram Ebrahimnejad, Majid Saeedi, Darius Armstrong-James, Mahdi Abastabar, Hamid Badali

**Affiliations:** 1Student Research Committee, Mazandaran University of Medical Sciences, Sari 48157-33971, Iran; robabebrahimibarogh@gmail.com (R.E.B.); abolfazlmaleki93@gmail.com (A.M.); fatemehkhosra@gmail.com (F.K.E.); zahra_yahyazade77@yahoo.com (Z.Y.); 2Invasive Fungi Research Center, Communicable Diseases Institute, Mazandaran University of Medical Sciences, Sari 48157-33971, Iran; imaan.haghani@gmail.com; 3Department of Medical Parasitology, Ardabil University of Medical Sciences, Ardabil 56189-53141, Iran; 4Pharmaceutical Sciences Research Centre, Haemoglobinopathy Institute, Mazandaran University of Medical Sciences, Sari 48157-33971, Iran; mobin.rahimnia@gmail.com (S.M.R.); pedram.ebrahimnejad@gmail.com (P.E.); majsaeedi@gmail.com (M.S.); 5Department of Pharmaceutics, Faculty of Pharmacy, Mazandaran University of Medical Sciences, Sari 48157-33971, Iran; 6Department of Laboratory Sciences, Sirjan School of Medical Sciences, Sirjan 76169-16338, Iran; nosratabadi.mohsen@yahoo.com; 7Department of Medical Mycology, School of Medicine, Mazandaran University of Medical Sciences, Sari 48157-33971, Iran; 8Department of Infectious Diseases, Medical Research Council Centre for Molecular Bacteriology and Infection, Imperial College London, London SW7 2AZ, UK; d.armstrong@imperial.ac.uk; 9Department of Molecular Microbiology & Immunology, South Texas Center for Emerging Infectious Diseases, The University of Texas at San Antonio, San Antonio, TX 78249, USA

**Keywords:** dermatophytosis, *Trichophyton indotineae*, guinea pig, luliconazole (LCZ), nanostructured lipid carriers (NLCs)

## Abstract

**Background:** *Trichophyton indotineae* terbinafine-resistant infections are emerging in healthy individuals. Luliconazole, an imidazole antifungal that is effective against skin infections, faces challenges due to low water solubility and poor skin penetration. This study aimed to formulate a luliconazole-loaded nanostructured lipid-carrier (NLC) gel in a Carbopol-based system to enhance drug absorption and efficacy in a guinea pig model of dermatophytosis. **Methods**: Luliconazole-loaded nanostructured lipid carriers (NLCs) were prepared using a solvent evaporation method and gel formulation. Skin absorption and retention were assessed via Franz diffusion cells. The antifungal efficacy was tested against *T. indotineae* in thirty guinea pigs with induced tinea corporis, divided into five treatment groups. Mycological, clinical, and histopathological evaluations were conducted, along with skin irritation studies for safety. **Results**: LCZ-NLC demonstrated significantly better skin penetration than simple luliconazole gel, with cumulative drug penetration of 71.8 ± 3.7 μg/cm^2^ versus 50.9 ± 4.2 μg/cm^2^ after 24 h. Both formulations achieved complete infection resolution after 21 and 28 days, with reduced inflammation and no local irritations. On day 21, the LCZ-NLC 1% gel significantly reduced lesion scores and mycological evidence of infection compared to the terbinafine-treated groups, untreated controls, and NLC-gel-treated group (*p* < 0.05). Histopathological analysis indicated a reduction in both epidermal thickening and fungal burden in the models that received treatment with the LCZ-NLC 1% gel. **Conclusions**: Luliconazole-loaded lipid carriers enhance drug absorption and efficacy, suggesting shorter treatment durations and improved patient outcomes for resistant fungal infections. However, further studies are warranted to correlate these findings with clinical outcomes.

## 1. Introduction

Dermatophytes are a significant cause of dermatophytosis, a common fungal infection that affects approximately 20–25% of the global population [1,2]. The rising incidence of dermatophytosis infections can be attributed to several factors, including insufficient hygiene practices, socioeconomic changes, lifestyle-related diseases such as diabetes, animal breeding, improper antibiotic use, excessive corticosteroid use, and the emergence of antifungal resistance [3]. Among dermatophytes, the *Trichophyton* genus, particularly *Trichophyton indotineae* (formerly, *T. mentagrophytes* genotype VIII), is highly contagious and responsible for chronic or recurrent skin, hair, and nail infections. Symptoms include itching, inflammation, and extensive body wounds, which pose significant treatment challenges [4,5,6,7]. Terbinafine (TRB) is a widely prescribed antifungal medication for dermatophyte infections; however, resistance to this drug is primarily attributed to point mutations in the SQLE gene, which is crucial for ergosterol production [8]. Consequently, accurate diagnosis and effective treatment of dermatophytosis remain significant challenges. Systemic therapies are increasingly necessary, including itraconazole and newer agents like voriconazole and luliconazole (LCZ) [9]. LCZ, an FDA-approved imidazole, demonstrates remarkable efficacy against a range of fungi, including dermatophytes and *Candida* spp., *Aspergillus* spp., *Fusarium* spp., *Madurella* spp. (including *M. mycetomatis*, *M. pseudomycetomatis*, *M. tropicana*, and *M. fahalii*), *Falciformispora senegalensis*, *Medicopsis romeroi*, *Trematosphaeria grisea*, and *Leishmania* secies [10,11,12,13,14,15]. This medication is typically applied as a 1% cream and a 10% solution for the treatment of fungal infections. LCZ easily penetrates the nail plate and skin, quickly reaching the deepest level of fungal infection [16]. Recent advancements in nanotechnology have introduced nanostructured lipid carriers (NLCs) as promising drug delivery systems. These carriers enhance dermal drug absorption and minimize toxicity or sensitivity, making them suitable for treating skin diseases [17]. The small size of these nanoparticles, when used in dermal drug delivery, comes into direct contact with the stratum corneum, increasing skin moisture and transferring the loaded drug to the underlying layers of the skin [18]. These carriers have been used in many studies in injectable [19], inhalable [20], oral [21], ocular [22], and topical drug delivery [23]. While NLCs have been widely studied globally for various drug delivery systems, our study investigates their application for LCZ explicitly, filling a significant gap in the literature. This research is particularly relevant as it explores the potential of NLC in enhancing the therapeutic efficacy of LCZ in animal models, which has not been previously addressed. This study aims to evaluate a Luliconazole-loaded nanostructured lipid carrier (LCZ-NLCs) gel, synthesized in a Carbopol-based system, to reduce side effects, enhance efficacy, and achieve controlled release in a guinea pig model of dermatophytosis [24]. This research can potentially improve therapeutic outcomes for patients suffering from dermatophyte infections by addressing the limitations of current antifungal treatments and exploring the potential of NLCs for luliconazole delivery. The results could enhance the scientific understanding of dermatophytosis treatment and have considerable implications for public health and clinical practice, potentially decreasing these infections’ morbidity and financial burden.

## 2. Materials and Methods

### 2.1. Preparation of LCZ-NLC NPs

The LCZ-NLCs were synthesized and well-characterized in our previous study [24]. The process involved dissolving LCZ, Span 80, Glycerol Monostearate (GMS), and oleic acid in chloroform, forming the organic phase. A heated aqueous phase containing Tween 80 was then added dropwise to the organic phase using a high-shear homogenizer. The resulting emulsion was sonicated and cooled in an ice bath to produce LCZ-NLCs. Finally, the mixture was stirred overnight at room temperature to remove any residual organic solvent.

### 2.2. Gel Preparation

The formulated mixture was converted to lyophilized powder after a freeze-drying cycle. It was then re-dispersed in a specified volume of deionized water containing a preservative to achieve a drug concentration of 1% *w*/*v* in the gel. Subsequently, 1% *w*/*v* of Carbopol 941 was added and left for several hours to fully hydrate the Carbopol. Finally, triethanolamine neutralized the Carbopol, forming a gel network [25]. The control gel was prepared similarly, with 1% *w*/*v* of LCZ dispersed in deionized water and a drop of Tween 80. Carbopol and triethanolamine were then added, resulting in the creation of the LCZ simple gel.

### 2.3. In Vitro Drug Skin Absorption and Retention

This experiment used pieces of skin from male guinea pigs weighing 370 to 550 g. A mixture of 13 mg xylazine/kg and 87 mg ketamine/kg body weight was utilized to anesthetize the guinea pigs, who were then euthanized by inhaling chloroform. The abdominal skin was shaved using an electric trimmer and surgically removed, and the fatty tissues were separated. The skin samples were immediately preserved in a standard saline solution for 24 h at 4 °C. After 24 h, the skins were placed in a Franz cell with an effective diffusion area of 3.8 cm^2^, and the removed skins were fixed between the donor and receiver compartments using adhesive, allowing contact with the test solution [26]. The receiver compartment was filled with water and ethanol (50:50), and the cells were maintained at a temperature of 32 ± 0.5 °C, controlled by a thermostatic system with water circulating around the cell. The LCZ simple gel served as a control, while the LCZ-NLC gel was applied evenly to the dorsal skin in the Franz cell. Samples were taken from the receiver environment at specified intervals (1, 2, 4, 6, 8, 10, and 24 h) [27]. The collected samples were analyzed using a UV–Vis spectrophotometer at 298 nm to determine the drug content in the obtained samples. After the 24 h test, the skins were removed from the Franz cell, washed with normal saline, and cut into small pieces with scissors. These pieces were placed in a Falcon tube to be immersed in 50% ethanol. Finally, the skin pieces were placed in a sonic bath for 30 min to extract LCZ from the skin and determine the remaining drug in the skin layers. The resulting solution was filtered and analyzed at 298 nm.

### 2.4. Selection of Fungi

This study utilized a terbinafine-resistant *T. indotineae* isolate (OQ214848) from the Sample Bank at the Invasive Fungal Research Center. The isolate was identified through Sanger sequencing of the ITS region. The isolate exhibited two specific point mutations: Phe397Leu (F397L) and Ala448Thr (A448T), which result in the substitution of single amino acids at critical positions in the SQLE protein, thereby contributing to its resistance [28,29].

### 2.5. Preparation of Dermatophyte Suspension for Animal Inoculation

To prepare a fungal suspension, dermatophytes are cultured in potato dextrose agar (PDA, Difco) medium and kept for two weeks at 30 °C until sporulation occurs. A thermal shock was performed to induce sporulation. Then, a sterile swab soaked with distilled water was stretched over the culture medium, and some spores were removed. After 5 s of vortexing, 5 to 10 min were given for the fungal hyphae to settle. The supernatant was removed from the solution, and the conidia were counted with a Neubauer slide to reach the required concentration of 1 *×* 10^8^ cells/mL.

### 2.6. In Vivo Animal Study

This experimental study involved 30 healthy five-month-old male guinea pigs, also weighing 370 to 550 g, used for the in vivo assessment of drug absorption and retention [30]. The guinea pigs were housed independently in a completely clean and disinfected cage at a temperature of 25–27 °C, relative humidity of 50 ± 20%, and lighting 12 h per day with ventilation 12–15 times per hour via an all-fresh air system [31]. The care and usage of the animals was conducted in accordance with the legal criteria for the humane treatment and management of animals (The Act on Welfare and Management of Animals [Act No. 105 of 1973, Act No. 68 of 2005], Japan]) and met the standards set by the institute’s animal welfare committee [32]. In this research, the tested animals were randomly divided into 5 groups, each containing 6 guinea pigs: 2 main treatment groups (treatment with LCZ topical 1% gel, treatment with LCZ-loaded NLC topical gel), a negative control group (NLC topical gel), a group treated with terbinafine, and a positive control group (without the use of antifungal drugs).

### 2.7. Preparation Method of Tinea Corporis in Guinea Pigs

A combination of ketamine (80 mg/kg of animal body weight) and xylazine (8–10 mg/kg of animal body weight) was used to anesthetize the animals [33]. Using clean and sharp scissors, the hair on the back, near the neck, of all the piglets was cut to a 2.5 × 2.5 cm^2^ size. Then, the area was shaved with a razor, making the skin completely hair-free. After that, several scratches were made on the skin using a sharp, sterile scalpel. Finally, the skin was inoculated with a 100 μL solution containing 1 × 10^8^ cells/mL of resistant *T. indotineae* (18).

### 2.8. Evaluation of Treatment

The tinea corporis animal model was used to evaluate the efficacy of LCZ and LCZ-loaded NLC gel as antifungal agents. Topical antifungal agents were used when the primary lesion appeared in each animal model. In our cases, treatment started 10 days after infection onset. The drugs were applied topically to the skin twice daily for 28 days (1 g per site) [34].

### 2.9. Mycological Examination

Mycological examinations were conducted on days 10, 17, 24, 31, and 38 after inoculation to confirm the establishment of dermatophytes and the treatment of dermatophytosis in the sampling area. Skin and hair were collected for direct microscopic examination, with 20% potassium. Skin samples were cultured on Sabouraud dextrose agar with chloramphenicol and cycloheximide (SCC, Difco) and incubated at 27 °C for 14 days. Skin biopsies were taken before and after treatment and stained with hematoxylin–eosin for skin inflammatory disorders. The clinical evaluation consisted of a semi-quantitative score in which inoculated samples were evaluated and compared with uninfected areas taken from the same animal. The following scores were assigned: 0 (no lesion), 1 (hair loss only), 2 (redness with scaling), 3 (significant redness with extensive scaling), 4 (ulcers and scars with symptoms), and 5 (severe skin lesions, redness, flaking, and lack of hair growth). Moderate scores indicate degrees of severity [35]. The percentage of efficacy was calculated using the following equation:

Percent efficacy = 100 − (T × 100/C)% where T is the total score of the treatment group and C is the total score of the untreated control. The total score for any group denotes the average clinical score of five different animals in the same group.

### 2.10. Histopathology Analysis

Skin biopsy samples were obtained from one animal per group at both the beginning and end of the study for histopathological examination. A 3 mm diameter piece of skin was collected from an anesthetized animal in each group using a disposable sterile dermal biopsy punch. The collected tissue was fixed in 10% neutral buffered formalin, then embedded in paraffin and processed for histopathological analysis using PAS (Periodic Acid-Schiff) and H&E (hematoxylin–eosin) staining (Chemi pazhooh Asia, Iran). Although H&E staining was used in this study to assess overall histopathological characteristics, including inflammation and tissue response, it is important to note that PAS staining was generally preferred for visualizing fungal elements in our work using light microscopy.

### 2.11. In Vivo Skin Irritation Studies

This study aimed to evaluate the potential of LCZ-NLC gel and LCZ simple gel to induce skin irritation. The study involved two groups of male guinea pigs, each consisting of three guinea pigs. A square centimeter area measuring 2 × 2 cm^2^; on the dorsal surface of the body was demarcated, and the hair within that area was shaved off. The initial group was treated with LCZ-NLC gel, and the second group was treated with LCZ simple gel. Each guinea pig received a 1 g formulation dose based on their assigned group. At 24 and 72 h intervals, the skin was examined for signs of edema, erythema, or rashes [36,37].

### 2.12. Statistical Analysis 

The experimental data were analyzed using GraphPad Prism version 9.1.1. Statistical evaluations, including group comparisons, were conducted using Kruskal–Wallis and Dunn’s post hoc tests. 

## 3. Results

### 3.1. Analysis of Drug Absorption and Retention in Gel Formulations

According to Figure 1, in the nano lipid carrier gel formulation of LCZ, greater penetration into the skin layers was observed compared to the simple drug gel (*p* < 0.05). Specifically, the average cumulative drug permeation through the skin after 24 h for the LCZ-NLC gel was 71.8 ± 3.7 µg/cm^2^; meanwhile, for the LCZ simple gel, it was 50.9 ± 4.2 µg/cm^2^.

As shown in Figure 2, the amount of drug remaining in the skin significantly increased compared to the control samples (*p* < 0.05). The average cumulative drug amount in the skin after 24 h for the LCZ-NLC gel was 68.1 ± 5.7 µg/cm^2^; meanwhile, for the LCZ simple gel, it was 43.2 ± 2.6 µg/cm^2^.

### 3.2. Animals

In this study, 30 animals were infected with a suspension inoculation of drug-resistant *T. indotineae*. All animals had ringworms, typically characterized by a red, itchy, scaly, circular rash. Skin lesions from the affected areas were collected at five distinct times: on the tenth day following infection and the seventh, fourteenth, twenty-first, and twenty-eighth days following therapy.

### 3.3. Efficacy of LCZ-NLC 1% Gel, LCZ 1% Gel, and Terbinafine 1%: A Clinical and Mycological Evaluation in Guinea Pig Models

After a single inoculation of 1 × 10^8^ cells/mL of drug-resistant *T. indotineae*, an inflammatory response was observed at the inoculation site on day 10. From this day onward, local treatment with LCZ 1% and LCZ-loaded NLC gel was administered twice a day for 28 days. The positive reaction to *Trichophyton* indicated the establishment of an immune response to *T. indotineae*.

In assessing the mean score of clinical lesions in the group treated with NLC gel, a score of 3 (significant redness with extensive scaling and hair loss) was recorded in the first week of treatment. In the first week of treatment, the groups treated with LCZ 1% gel, LCZ-NLC 1% gel, terbinafine 1%, and untreated control showed scores of 3 and 4 (significant redness with extensive scaling, loss of hair, and ulcer). The decrease in lesion scores for the LCZ-NLC 1% gel- and LCZ 1% gel-treated groups occurred between weeks 2 and 3. The downward trend continued until day 21, and on day 28 of treatment initiation, these two groups’ scores were approximately zero. In the fourth week post-treatment, the terbinafine and control groups scored 4 (significant redness with extensive scaling and hair loss) (Figure 3). Figure 3 compares the guinea pig skin infection in photographs taken on day 10 of infection (before beginning treatment) and weekends 1, 2, 3, and 4, when the clinical efficacy test of the control and various drugs tested was conducted. The untreated control and positive control groups received 1 g (g) of terbinafine, and the NLC gel groups showed patches, hair loss, and readily visible scaly skin (Figure 3). In contrast, improvements in the appearance of the guinea pig skins with minimal hair loss and mild ulcers were noted in the LCZ 1% gel group at 3 weeks. By the fourth week, this group scored zero, indicating complete recovery with no signs of infection. Meanwhile, the group treated with the LCZ-NLC 1% gel demonstrated normal hair growth and showed no signs of infection at 3 and 4 weeks. Figure 3 displays the clinical efficacy scores of each tested drug in comparison to the infected and untreated control. A lower clinical score indicates that the treatment is more effective than no treatment (as evidenced by comparison with the untreated control group). The results show a significant decline in lesion scores for the LCZ 1% gel and LCZ-NLC 1% gel compared to the untreated control, terbinafine, and NLC groups (*p* < 0.05), underscoring the unwavering reliability and validity of these results. Moreover, the lesion score in the NLC gel and terbinafine groups did not differ significantly from the untreated control (*p* > 0.05). While LCZ-NLC 1% gel demonstrated greater efficacy than LCZ 1% gel, it is important to note that there was no statistically significant difference between them (*p* = 0.4295). The LCZ-NLC 1% gel group showed significant improvement against dermatophytosis caused by *T. indotineae*, confidently outperforming the positive control group receiving 1 g of terbinafine (Figure 4 and Figure 5).

Images of skin infected with *Trichophyton indotineae* in guinea pigs at various time points illustrate the healing process. In Figure 4, the group treated with LCZ 1% gel achieved complete recovery by day 28, while the LCZ-NLC 1% gel group demonstrated full recovery by day 21. In contrast, the NLC gel, terbinafine, and control groups showed no significant improvement during the same period, as depicted in Figure 5.

Mycological investigations into skin scrapings were conducted on the 10th day after inoculation, confirming all animals were mycologically positive. Direct microscopic examination using 10% KOH showed significant positivity in the untreated control group, indicating active infection. Treatment with NLC-LCZ 1% gel and LCZ 1% gel significantly reduced the number of positive specimens, both in direct examination and culture, by weeks 2 and 3, demonstrating 100% mycological efficacy. No colonies were recovered from the NLC-LCZ gel and LCZ-treated skin specimens in culture for at least 21 and 28 days. In contrast, terbinafine and NLC gel treatments did not decrease the number of positive specimens by day 21.

### 3.4. Histopathological Evaluation of Dermatophytosis in Guinea Pigs

Skin biopsies were subjected to histopathological analysis using H&E and PAS staining to assess changes in epidermal structure and levels of inflammation. Skin samples were collected at two key time points: before and after the treatment protocol’s completion. The results are presented in Figure 6 and Figure 7. In groups A and B, guinea pigs received different treatments: LCZ simple gel in group A and LCZ-NLC gel in group B. H&E staining showed significant thinning of the epidermis in both treatment groups compared to untreated controls (Figure 6A,B). Moreover, these groups demonstrated reduced inflammation severity in the dermal layers (Figure 7A,B). Skin samples from infected guinea pigs treated with NLC exhibited a spongiotic psoriasiform epidermis with a parakeratotic stratum corneum (Figure 6C). There was also prominent neutrophilic infiltration in the epidermis and a mixed inflammatory cell presence in the upper dermis (Figure 7C). Conversely, skin specimens from untreated guinea pigs in group D revealed even more excellent epidermal thickening and hyperkeratosis (Figure 6D). Additionally, PAS-stained sections from the same group (group D) showed fungal hyphae within the hyperkeratotic stratum corneum (Figure 6E). These histological changes across all groups were consistent with findings from direct KOH microscopy and fungal culture studies, underscoring the effects of the various treatments on skin pathology.

### 3.5. Skin Irritation Studies

Skin irritation was researched for the NLC loaded with LCZ gel and LCZ simple gel because topical formulations are not intended to cause skin irritation. With LCZ-NLC gel and LCZ simple gel, there was no sign of irritation for 24, 48, or 72 h, including erythema, edema, or rashes (Figure 8). Because NLCs are made of physiological lipids, they do not have any adverse side effects. Also, these lipids are broken down in the body into fatty acids and glycerol, which are the typical elements of the body [38].

## 4. Discussion

This research addresses the challenges of treating dermatophytosis, specifically focusing on issues such as treatment unresponsiveness and antifungal resistance [39]. As a result, the search for new and effective substances, such as LCZ, has become necessary. This study evaluates the effectiveness of a topical gel containing 1% LCZ-NLC in treating animal-origin dermatophytosis, using guinea pigs as the most sensitive laboratory animal. Niwano et al., Mahshid Lalv, and Hiroyasu Koga et al. also chose the guinea pig model as the most sensitive laboratory animal to investigate the extracorporeal effects of anti-dermatophytic activities, especially against *T. mentagrophytes* [35,40,41]. The skin inoculation method on the guinea pig selected only the infected area on the animal’s back. This area was selected because the animal cannot lick or clean the lesion site, has limited contact with hands and feet, and provides a visible surface for infection assessment [42,43]. The tinea corporis model effectively causes infection with one inoculation, making it a time-efficient method for testing antifungal drug activity. One can select the severity of the model used by choosing the appropriate number of days after inoculation for analysis because the inflammatory response at the infected site increases daily after the inoculation [31].

LCZ, a potent topical antifungal imidazole, is under clinical development in the United States for the short-term treatment of dermatophytosis. A study investigated the clinical benefits of LCZ cream in guinea pig models of tinea corporis and tinea pedis induced with *T. mentagrophytes*. Results showed that even at a concentration of 0.02%, LCZ cream improved skin lesions and eradicated fungus in half or less of the time required for 1% terbinafine cream and 1% bifonazole cream [31]. This medicine is a 1% cream and a 10% solution. LCZ easily penetrates the nail plate and skin and quickly reaches the highest fungicidal level [16]. However, despite its efficacy, the short duration of LCZ’s action may limit its effectiveness in prolonged treatment scenarios. This is where nanotechnology, specifically NLCs, becomes relevant. In recent years, nanotechnology has become increasingly prevalent in medicine and pharmaceuticals. It has proved to be successful in advanced drug formulation, targeted drug delivery, and controlled drug release. NLCs are a new generation of nanosystems for transporting drugs. They are composed of solid and liquid fats that can carry different drugs due to their unique structure, which is both lipophilic and hydrophilic [17]. The fats that make up the structure of these nanoparticles are highly suitable for delivering drugs to the skin. They are non-toxic and non-irritant, making them ideal for treating skin conditions. The small size of these nanoparticles allows them to come into direct contact with the outermost layer of the skin, known as the stratum corneum. This interaction forms a thin layer on the skin’s surface, increasing skin moisture. The drugs loaded into these nanoparticles are then released into the deeper layers of the skin [18]. These carriers have been used in many types of research in the field of injectable [19], inhalation [20], oral [21], ocular [22], and topical [23] drug delivery.

Nonionic surfactants are widely recognized for their low toxicity and minimal skin irritation, making them popular as skin penetration enhancers in various pharmaceutical formulations [44]. Their enhancement properties stem from both direct and indirect mechanisms. Directly, surfactants interact with the skin barrier, modifying its structure and facilitating permeant penetration. Indirectly, they influence the thermodynamic activity of the permeant within the delivery vehicle, enhancing the driving force for diffusion across the skin [45]. This dual action improves the efficacy of drug delivery systems and underscores the relevance of nonionic surfactants in the formulation of topical therapies. Furthermore, synergistic effects have been reported when combining specific binary mixtures of surfactants [46]. This synergistic effect may explain the enhanced skin penetration observed with LCZ-NLCs compared to simple LCZ gel in this study. In 2016, Kansagra and colleagues developed microemulsion-based gels (MBGs) for LCZ. They argued that the solubility of this drug in water is very low. They found that the new compound, evaluated with various oils, had increased effectiveness and reduced sensitivity [47]. Kumar and his colleagues designed a gel form of nano-crystal LCZ in 2019. They discovered that this medication can be trapped in the skin 88% more frequently than its raw form, making it safer [48]. We used the conventional tinea corporis model to test treatment efficacy with 1% LCZ and LCZ NLC gels. Then, we adopted the accurate tinea corporis model to further confirm the efficacy of therapy. The study used NLC gels as reference drugs. Both the LCZ and NLC-LCZ gels were effective in treating tinea corporis. After treatment with a 1% concentration of either gel, the lesions significantly improved. These results were consistent with those of previous randomized clinical trials, where LCZ gel (0.1–1%) exhibited significant antimycotic activity against tinea pedis at doses as low as 0.1%. Further research has been conducted to evaluate the effectiveness of LCZ on fungi. In 2021, Lalvand M. and colleagues conducted an in vivo study on 36 male Indian piglets with dermatophytosis. The use of topical cream containing 5% fluconazole and 1% terbinafine, along with nano-fluconazole and nano-terbinafine solutions, showed that treatment of dermatophytosis with drug nanoparticles increases the rate of lesion improvement [35]. The limitations of studies on topical medications include small sample sizes, a lack of standardization in application, and a lack of long-term follow-up data. Consider ethical implications that may arise during a study, such as adverse events or side effects that could potentially restrict the use of animal models.

## 5. Conclusions

Terbinafine-resistant dermatophytosis is increasingly common, highlighting the need for alternative treatment options. The NLC-LCZ 1% gel shows significant promise for managing resistant cases due to its enhanced antifungal properties. NLCs lead to improvements in skin permeation and prolong the drug retention of LCZ, making this formulation a viable alternative to traditional treatments. Continued research is essential to optimize the efficacy of NLC-LCZ gel and enhance clinical outcomes for patients suffering from resistant dermatophytosis.

## Figures and Tables

**Figure 1 jof-11-00324-f001:**
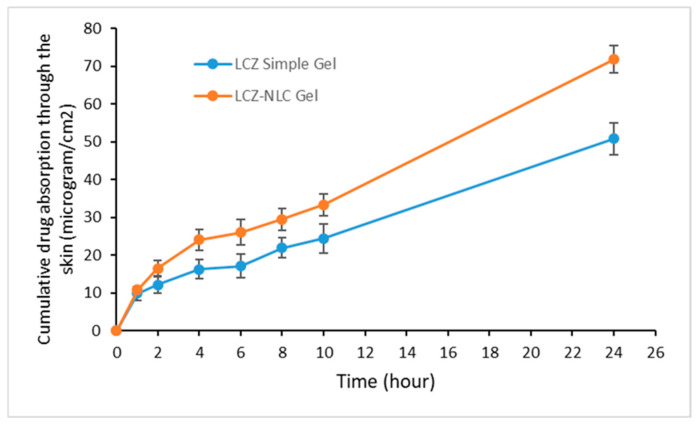
Comparison of the cumulative amount absorbed from the skin of guinea pigs in µg/cm^2^ for the nano lipid carrier of luliconazole formulation (LCZ-NLC gel) and the simple drug gel after 24 h. The data are presented as mean ± SD, n = 3. Statistical analysis using ANOVA indicated significant differences (*p* < 0.05).

**Figure 2 jof-11-00324-f002:**
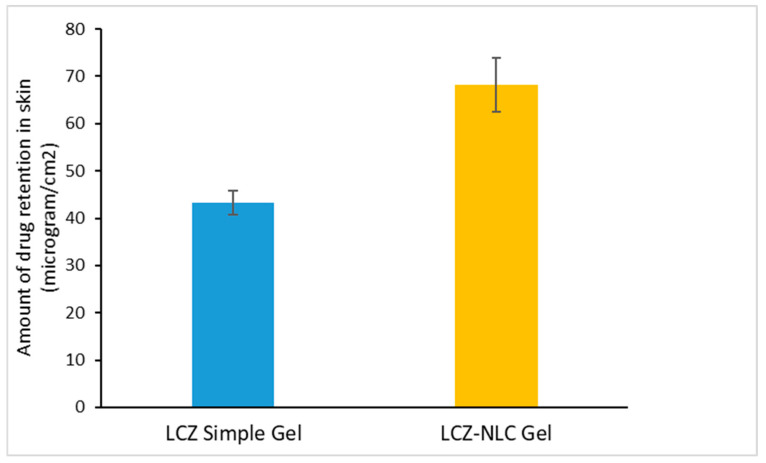
Comparison of drug retention between the nano lipid carrier of luliconazole gel (LCZ-NLC gel) and the simple drug gel in the skin layers of guinea pigs after 24 h. Data are presented as mean ± SD (n = 3). Statistical analysis using ANOVA indicated significant differences (*p* < 0.05).

**Figure 3 jof-11-00324-f003:**
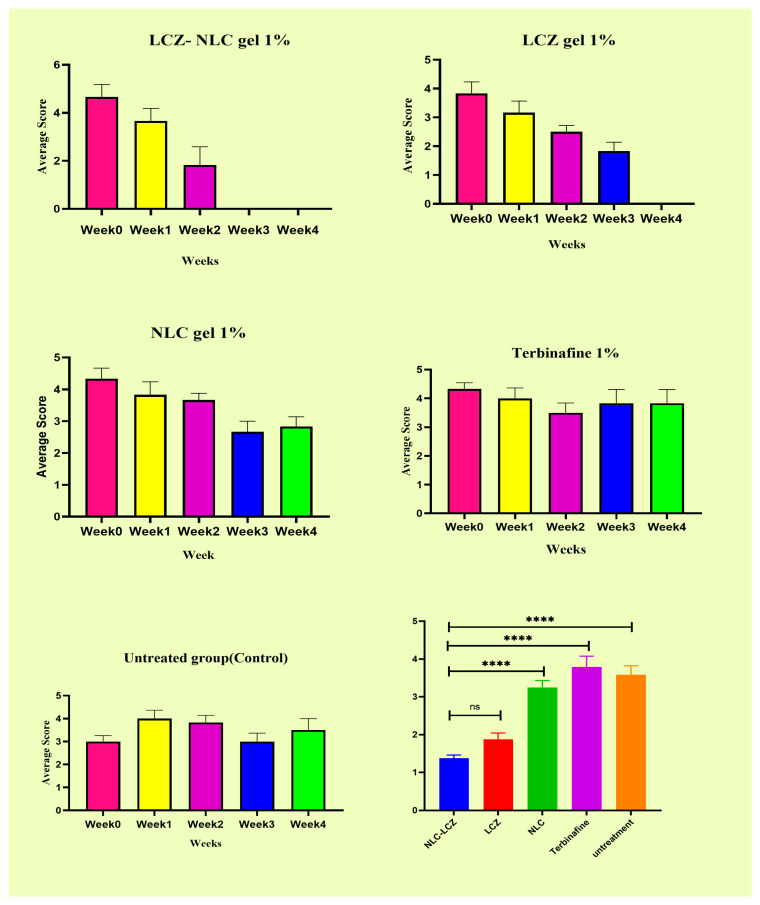
Lesion scores in guinea pigs infected with the *T. indotinea* strain. Average lesion scores are compared from week 0 (before treatment) to weeks 1, 2, 3, and 4 (after treatment initiation) across treatment groups: LCZ-NLC 1% gel, LCZ 1% gel, NLC 1% gel, terbinafine, and untreated controls. Statistical significance is indicated by asterisks (****: *p* < 0.0001; ns: not significant).

**Figure 4 jof-11-00324-f004:**
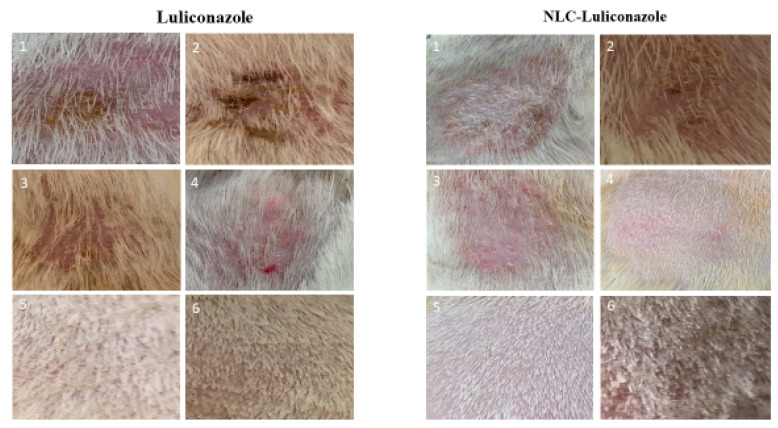
Images of skin infected with *Trichophyton indotineae* in guinea pigs at various time points: (1) 10 days post-infection (before treatment), (2) 7 days during treatment, (3) 14 days during treatment, (4) 21 days during treatment, (5) 28 days during treatment, and (6) 35 days post-treatment.

**Figure 5 jof-11-00324-f005:**
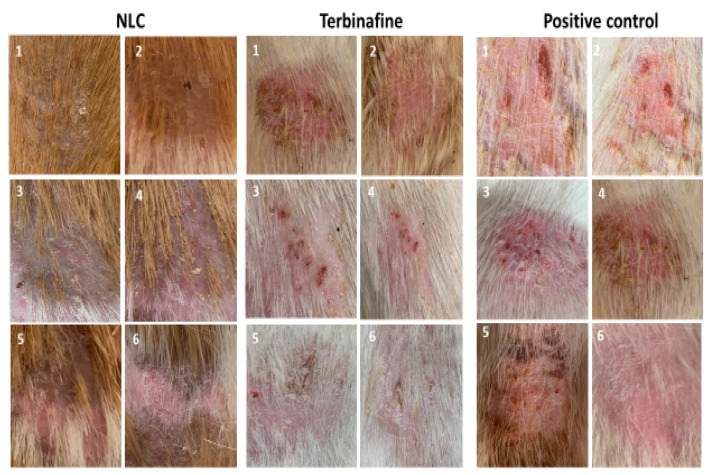
Images of skin infected with *Trichophyton indotineae* in guinea pigs before and after treatment with plain NLC gel and terbinafine, along with a positive control. The images illustrate the condition of the skin at various time points: (1) 10 days post-infection (before treatment), (2) 7 days during treatment, (3) 14 days during treatment, (4) 21 days during treatment, (5) 28 days during treatment, and (6) 35 days post-treatment.

**Figure 6 jof-11-00324-f006:**
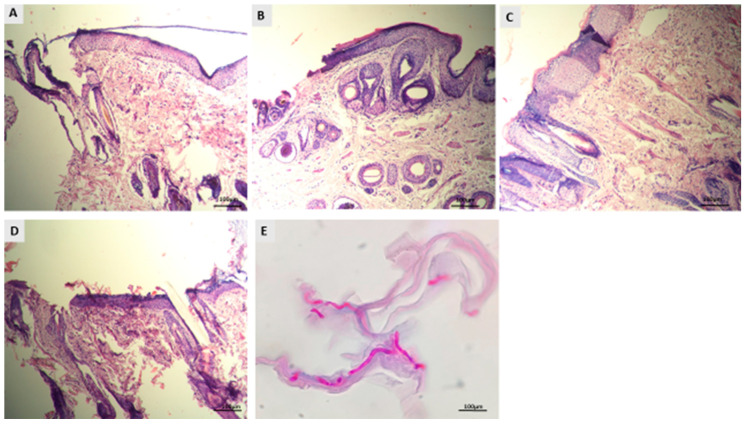
Histologic features of the guinea pig model of dermatophytosis following treatment with LCZ simple gel, LCZ-NLC gel, NLC gel (placebo), terbinafine cream, and untreated controls. Images were taken 10 days post-inoculation ((**D**) untreated controls) and four weeks after treatment (**A**–**C**). (**A**) Group A after LCZ simple gel, (**B**) group B after LCZ-NLC gel, (**C**) group C after NLC gel, and (**D**) untreated controls. (**E**) PAS-stained sections show fungal hyphae in the hyperkeratotic stratum corneum. Groups A and B had a thinner epidermis than controls, while groups C and D exhibited a thicker epidermis with increased inflammatory infiltration. All images were taken at a magnification of 40×.

**Figure 7 jof-11-00324-f007:**
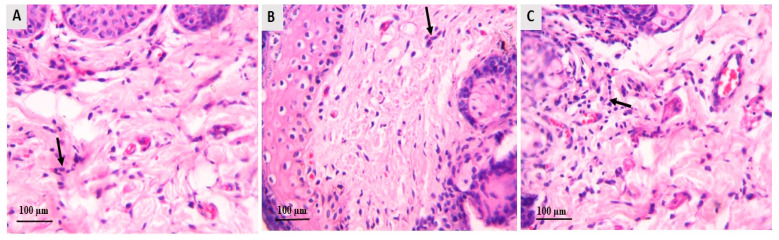
Histologic features of the guinea pig model of dermatophytosis following treatment with LCZ simple gel, LCZ-NLC gel, NLC gel (placebo), terbinafine cream, and untreated controls. Images depict (**A**) infected group A after LCZ simple gel, (**B**) group B after LCZ-NLC gel, and (**C**) group C after NLC gel treatment, taken four weeks after treatment. Groups treated with luliconazole (LCZ simple gel and LCZ-NLC gel) showed a notable reduction in inflammation and inflammatory cell infiltration in the dermal layers, indicated by black arrows. In contrast, the NLC-gel-treated group exhibited significant inflammatory cell infiltration, also highlighted by black arrows. All images were captured at a magnification of 40×.

**Figure 8 jof-11-00324-f008:**
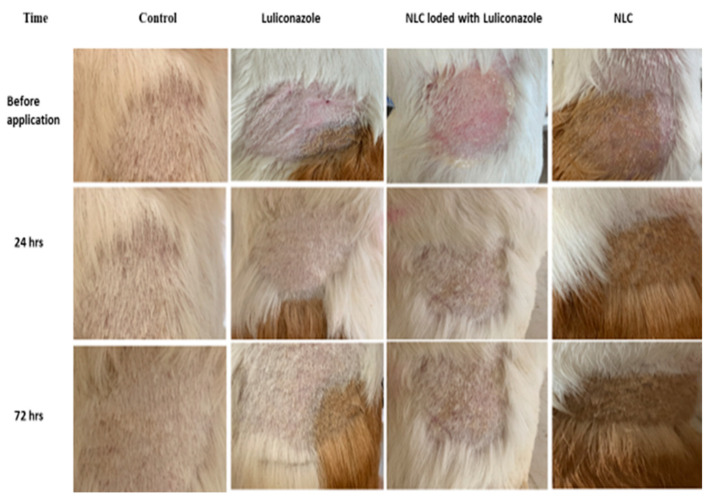
In vivo skin irritation test on Guinea pigs’ dorsal body surface. *n* = 3.

## Data Availability

The original contributions presented in this study are included in the article. Further inquiries can be directed to the corresponding authors.

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
