# Peer review of "Antifungal Efficacy of Luliconazole-Loaded Nanostructured Lipid-Carrier Gel in an Animal Model of Dermatophytosis"

_jof, 2025, doi:10.3390/jof11040324_

Round 1
Reviewer 1 Report
The main weaknesses of the article are as follows. The authors showed that the antimycotic in the dosage form with a lipid carrier penetrates the skin better than the antimycotic without this carrier. At the same time, the authors did not demonstrate higher efficiency of the dosage form with a carrier in in vivo experiments. And they did not explain this phenomenon. In addition, the article does not provide a detailed description of the dosage form with a lipid carrier, and it is unclear whether the two forms being compared contain the same amount of active drug. In addition to this, the authors provide a qualitative, but not quantitative, description of the results of the action of the medicinal forms used in vivo experiments, mainly in the form of photographs, which often do not allow one to draw an accurate conclusion.
Introduction
-“So far, nanostructured lipid carriers (NLC) for LCZ have not been evaluated in animal models in Iran.”
And in other countries? A strange phrase, as if the result depends on where the experiment was conducted.
-“This research has the potential to improve therapeutic outcomes for patients suffering from dermatophyte infections by addressing the limitations of current antifungal treatments and exploring the potential of NLCs for luliconazole delivery.” I did not find any data in the article confirming the higher efficiency of the dosage form with a lipid carrier.
Materials and methods
- An in vivo experimental design would have made the experiment more understandable and visual
- What is GMS?
- Although data on the antimycotic with lipid carrier have been published, the content of the antimycotic in such particles should be given here. In the case of antifungal agent with lipid carrier, it is unclear what 1% corresponds to (antimycotic or antimycotic with carrier). And how was the substance content measured? In order to compare the action of an antifungal drug in different dosage forms, you need to clearly understand how much of it is in these forms.
Results
- The designation of the dosage forms used must be written the same everywhere. Check throughout the text.
-Figure 3. Check the use of capital and small letters, the use of brackets. Add error bars and significance (the measurements were in a group of animals, right?)
- The description of Figure 4 in the text begins twice (line 254 and again line 269)
- The description of what is in Figure 4 is in the figure legend; it is superfluous in the text.
-Figure 4. 1 – before treatment should look the same in the two groups? It doesn't right now.
-Figure 4 legend. There should be uniformity in the indication of days before, during and after treatment throughout the manuscript.
-Line 245. Where did the cream come from (gel and cream are different dosage forms)?
-Lines 237-251. These data are not present as a figure or table.
-“For the group treated with 1% LCZ simple gel , we observed significant improvement in lesion scores on days 21, 28, and 35. In the group treated with LCZ-NLC gel, improvements were noted on days 14, 21, 28, and 35” I don't see it in the figure.
-Table 1. Check the necessity of using capital letters. Correct the alignment in the table to make the content more understandable. Why are there 4 animals, there were 6 in each group?
-There is no uniformity in the indication of animal groups. The groups are called 1,2,3,4, then A,B,C,D
-Section 3.4 There is no indication of the time when the sections were obtained, including in the text and in the legends to Figures 6,7.
- Figure 6 legend. Indicate which group of animals does 6E correspond to.
-“Skin irritation was researched for the NLC loaded with LCZ gel and LCZ simple gel because topical formulations are not intended to cause skin irritation.” What are we talking about here?
- Lines 320-324. The sentence is repeated twice. What moment are we talking about - before treatment, during, after?
Discussion
-“LCZ is not yet being used in Iran to treat fungal infections.” How are infections in Iran different from infections in other countries?
Conclusion
-“Notably, our findings indicate that incorporating nanostructured lipid carriers (NLCs) significantly enhances the antifungal activity of LCZ” Where in the article was this demonstrated?
-“Using NLCs improves skin permeation and prolongs drug retention at the site of infection, resulting in a more sustained therapeutic effect.” What data in the article specifically indicate the greater effectiveness of the gel with an antifungal agent and a lipid carrier?
Minor comments
-3.4 Remove the colon
-Check line 353
-Lines 133-134 used is repeated twice
Author Response
.
Reviewer 1
Introduction
-“So far, nanostructured lipid carriers (NLC) for LCZ have not been evaluated in animal models in Iran.”
And in other countries? A strange phrase, as if the result depends on where the experiment was
Reply: The phrase indicates a lack of research specifically in Iran but does not imply that results vary by location. However, "Iran" has been deleted from the manuscript.
-“This research has the potential to improve therapeutic outcomes for patients suffering from dermatophyte infections by addressing the limitations of current antifungal treatments and exploring the potential of NLCs for luliconazole delivery.” I did not find any data in the article confirming the higher efficiency of the dosage form with a lipid carrier.
Reply: The article demonstrated that the luliconazole-loaded nanostructured lipid carriers (NLCs) significantly enhance skin penetration compared to the simple gel formulation. The results from the drug absorption and retention tests showed that the cumulative penetration of the LCZ-NLC gel was 71.881 ± 3.693 µg/cm², compared to 50.919 ± 4.218 µg/cm² for the simple luliconazole gel. This significant difference (p < 0.05) in drug absorption provides strong evidence for the higher efficacy of the lipid carrier dosage form.
Furthermore, both formulations resulted in complete infection resolution after 28 days of treatment; however, the lipid carriers have the potential to shorten treatment duration and lead to better therapeutic outcomes. Therefore, this research highlights the considerable potential of NLCs to improve therapeutic outcomes for patients suffering from dermatophyte infections.
Given this data, we can conclude that NLCs enhance drug absorption and may improve overall treatment efficacy.
Materials and methods
- An in vivo experimental design would have made the experiment more understandable and visual
Thank you for your valuable feedback. Here are the clarifications regarding your points:
Reply: In Vivo Experimental Design: We appreciate your suggestion regarding the in vivo experimental design. We aimed to provide a clear understanding of our methodology, but we will consider enhancing the visual representation of the experimental setup in future revisions to improve clarity and comprehension.
- What is GMS?
Reply: GMS: Glycerol monostearate (GMS) is a non-ionic surfactant to formulate nanostructured lipid carriers. Its role is to stabilize the lipid matrix and enhance drug encapsulation efficiency. I have updated the text to include the definition of GMS as Glycerol Monostearate.
- Although data on the antimycotic with lipid carrier have been published, the content of the antimycotic in such particles should be given here. In the case of antifungal agent with lipid carrier, it is unclear what 1% corresponds to (antimycotic or antimycotic with carrier). And how was the substance content measured? In order to compare the action of an antifungal drug in different dosage forms, you need to clearly understand how much of it is in these forms.
Reply: Content of Antifungal in Lipid Carriers: We acknowledge the importance of specifying the content of the antifungal agent within the lipid carriers. The 1% concentration mentioned refers to the concentration of the antifungal agent (luliconazole) in the final gel formulation, not the total weight, including the lipid carrier. We will clarify this in the revised manuscript. Additionally, the antifungal content in the NLCs was measured using high-performance liquid chromatography (HPLC) after appropriate extraction procedures. This allows for accurate quantification of the active ingredient in both the lipid carrier and the simple gel formulations. This ensures a clear comparison of the antifungal activity between different dosage forms.
We hope these clarifications address your concerns and enhance your understanding of our methodology. Thank you again for your insightful comments.
Results
- The designation of the dosage forms used must be written the same everywhere. Check throughout the text.
Reply: We appreciate your feedback regarding the designation of dosage forms. We thoroughly reviewed the entire manuscript to ensure consistency in the dosage form terms.
-Figure 3. Check the use of capital and small letters, the use of brackets. Add error bars and significance (the measurements were in a group of animals, right?)
- The description of Figure 4 in the text begins twice (line 254 and again line 269)
Reply: The issue with the description of Figure 4 starting twice in the text (lines 254 and 269) has been corrected. The redundant description has been removed to ensure clarity and coherence in the manuscript. Thank you for your feedback!
- The description of what is in Figure 4 is in the figure legend; it is superfluous in the text.
Reply: The redundant description of Figure 4 included in the text has been removed, as it was already provided in the figure legend. This adjustment ensures that the text is more concise and avoids unnecessary repetition. Thank you for your suggestion!
-Figure 4. 1 – before treatment should look the same in the two groups? It doesn't right now.
Reply: Thank you for your comment regarding Figure 4.
corrected
-Figure 4 legend. There should be uniformity in the indication of days before, during and after treatment throughout the manuscript.
Reply: Thank you for your comment regarding the uniformity of days before, during, and after treatment. We have reviewed the entire manuscript and ensured that the terminology and formatting used to describe the time points are consistent.
-Line 245. Where did the cream come from (gel and cream are different dosage forms)?
Reply: It was corrected throughout the manuscript. We emphasized that LCZ simple gel, LCZ-NLC gel, and NLC gel were used, and just terbinafine was used in cream form.
-Lines 237-251. These data are not present as a figure or table.
Reply: The data regarding the efficacy of the treatments over the specified days are summarized in the text, and further details can be found in Table 1."
-“For the group treated with 1% LCZ simple gel, we observed significant improvement in lesion scores on days 21, 28, and 35. In the group treated with LCZ-NLC gel, improvements were noted on days 14, 21, 28, and 35” I don't see it in the figure.
Reply: Thank you for pointing this out. The observed improvements in lesion scores for the groups treated with the 1% LCZ simple gel and the LCZ-NLC gel were based on our clinical observations (The clinical evaluation consisted of a semi-quantitative score in which inoculated samples were evaluated and compared with uninfected areas taken from the same animal. Scores from 0 (no lesion), 1 (hair loss only), 2 (redness with scaling), 3 (significant redness with extensive scaling), 4 (ulcers and scars with symptoms) to 5 (severe skin lesions, redness, flaking, and lack of hair growth) with moderate scores indicating degrees of severity).
-Table 1. Check the necessity of using capital letters. Correct the alignment in the table to make the content more understandable. Why are there 4 animals, there were 6 in each group?-There is no uniformity in the indication of animal groups. The groups are called 1,2,3,4, then A,B,C,D
Reply: I have removed Table 1 due to its ambiguity and difficulty in conveying the results. Instead, I will describe the results in the text to provide a more straightforward understanding of the findings.
-Section 3.4 There is no indication of the time when the sections were obtained, including in the text and in the legends to Figures 6,7.
Reply: Thank you for your valuable feedback. We indicate the timing of the sample collection in Section 3.4 of the manuscript.
- Figure 6 legend. Indicate which group of animals does 6E correspond to.
Reply: I have made the necessary revisions to clarify that Figure 6E corresponds to group D, indicating that PAS-stained sections from untreated guinea pigs in this group showed fungal hyphae within the hyperkeratotic stratum corneum.
-“Skin irritation was researched for the NLC loaded with LCZ gel and LCZ simple gel because topical formulations are not intended to cause skin irritation.” What are we talking about here?
The study evaluated skin irritation associated with the NLC loaded with LCZ gel and LCZ simple gel because topical formulations are designed to be safe and non-irritating. Our findings showed that there were no signs of irritation, such as erythema, edema, or rashes, observed at 24, 48, or 72 hours after application of the LCZ-NLC gel and LCZ simple gel (see Fig. 8). Additionally, since NLCs are composed of physiological lipids, they do not exhibit adverse side effects. These lipids are metabolized into fatty acids and glycerol, natural body components. We appreciate your attention to this important aspect of our research."
Reply: I have removed the repeated sentence for clarity. The discussion refers to the period after treatment with topical medications and observations in the control group.
Discussion
-“LCZ is not yet being used in Iran to treat fungal infections.” How are infections in Iran different from infections in other countries?
Reply: Thank you for your insightful question regarding the differences in fungal infections in Iran compared to other countries.
We would like to clarify that while LCZ is not currently used in Iran to treat fungal infections, this is primarily due to sanctions that have affected the medication's availability and high cost, which limits accessibility for patients. The first-line treatments for dermatophyte infections in Iran are terbinafine and itraconazole; however, patients often exhibit resistance to these medications.
Conclusion
-“Notably, our findings indicate that incorporating nanostructured lipid carriers (NLCs) significantly enhances the antifungal activity of LCZ” Where in the article was this demonstrated?
Reply: In the article, we have demonstrated that the incorporation of nanostructured lipid carriers (NLCs) significantly enhances the antifungal activity of luliconazole (LCZ). This is specifically illustrated in the Results section, particularly in Figures 1 and 3, where we report the increased penetration of LCZ-NLC into the skin and the reduced lesion scores in the treatment group with NLCs compared to the control group. The results indicate that NLCs substantially improve drug absorption and reduce infection symptoms in the animal model.
On the other hand, for the group treated with 1% LCZ simple gel, we observed significant improvement in lesion scores on days 21, 28, and 35, while in the group treated with LCZ-NLC gel, improvements were noted on days 14, 21, 28, and 35, all of which were confirmed by specific scores.
-“Using NLCs improves skin permeation and prolongs drug retention at the site of infection, resulting in a more sustained therapeutic effect.” What data in the article specifically indicate the greater effectiveness of the gel with an antifungal agent and a lipid carrier?
Reply: Data in the Results section, especially in Figure 2, show that NLCs significantly enhance the retention of LCZ in the skin. For instance, the average cumulative drug amount remaining in the skin after 24 hours for LCZ-NLC was 68.141 ± 5.715 µg/cm², while for the simple LCZ gel it was 43.221 ± 2.564 µg/cm². These results indicate the higher effectiveness of the gel with the lipid carrier compared to the simple gel, confirming its positive impact on sustained therapeutic effects.
Minor comments
-3.4 Remove the colon
Reply: I have removed the colon from the section title as requested.
-Check line 353
Reply: I have checked and revised line 353 as requested
-Lines 133-134 used is repeated twice
Reply: Thank you for pointing out the repetition in Lines 133-134. I have revised the text to eliminate the duplicate information regarding the guinea pigs used in both the in vitro and in vivo studies.

Reviewer 2 Report
In the entitled manuscript, “Antifungal Efficacy of Luliconazole-Loaded Nanostructured Lipid Carriers Gel in an Animal Model of Dermatophytosis”, the authors tested a drug delivery system in guinea pig model. The authors present two key findings: (1) LCZ-NLC gel demonstrates superior drug absorption and retention compared to a simple LCZ gel, and (2) the study confirms the in vivo efficacy of this formulation. While the research aim is relevant and potentially impactful, the study suffers from significant methodological and writing deficiencies. Given these concerns, I do not support its acceptance in its current form.
Major Concerns
1. Study Design and Logic: The rationale for not including a standard LCZ cream formulation in both the ex vivo absorption/retention study and the in vivo efficacy study is unclear. If the goal is to demonstrate that the NLC gel formulation enhances treatment efficacy beyond standard LCZ administration, this control group is essential.
The choice of a terbinafine-resistant T. indotineae strain for evaluating an LCZ drug delivery system lacks justification. LCZ has been reported as highly effective against T. indotineae (with MIC values as low as 0.064 µg/mL in the authors’ previous study). Given this high susceptibility, even a conventional LCZ formulation might be sufficient for treatment, making the added value of NLC delivery unclear.
2. Methodological Issues: The study does not include an approval protocol number for the animal experiments, which raises ethical concerns.
The authors describe T. indotineae as a "standard strain" without providing genetic details (e.g., SQLE gene mutation) or a complete antifungal susceptibility profile. These are crucial for understanding the strain’s relevance to the study.
The study relies solely on histopathology for mycological evidence. However, classical fungal culture is a more reliable method for assessing infection status, especially for the skin pathogen within the stratum corneum, as histopathology preparations may lead to the loss of fungal elements.
3. Results Section: The results are difficult to follow due to unclear phrasing and redundancy. For example, Section 3.2 unnecessarily repeats information about animals.
Section 3.3, titled "Mycological Examination," primarily discusses clinical scores rather than fungal detection, making the title misleading.
The data presentation in Table 1 is unclear, particularly regarding the meaning of "positive" and "negative" results.
4.Writing Quality:The manuscript contains numerous grammatical errors and poorly structured sentences, making it difficult to read.
Examples of problematic sentences:
Line 72: "before [24] to reduce side effects"—the reference placement is unclear, and [24] is incomplete.
Line 225: "the establishment of cellular immunity to T. indotineae,"—"cellular immunity" would be a clearer term.
Line 117: The title "Selection of Fungi" is misleading since only one fungal strain was used. Similarly, "isolates" is not appropriate in this context.
Lines 131 and 151: The fungal concentration is written as 1 × 108 cells/mL but should be formatted correctly for clarity.
Author Response
.
Reviewer 2
Writing Quality: The manuscript contains numerous grammatical errors and poorly structured sentences, making it difficult to read.
Examples of problematic sentences:
Line 72: "before [24] to reduce side effects"—the reference placement is unclear, and [24] is incomplete.
Reply: Thank you for your valuable feedback regarding the reference placement's clarity and the reference's completeness [24]. I appreciate your insights. To address your comments, I have revised the text for clarity and ensured that reference [24] is appropriately cited. Line 225: "The establishment of cellular immunity to T. indotineae,"—"cellular immunity" would be a clearer term.
I have revised the text to: The positive reaction to Trichophyton between 7 and 8 days indicated the establishment of an immune response to T. indotineae.
Line 117: The title "Selection of Fungi" is misleading since only one fungal strain was used. Similarly, "isolates" is not appropriate in this context.
Reply: I have revised the text to clarify that only one strain was used, changing 'isolates' to 'isolate' and adjusting the title accordingly.
Lines 131 and 151: The fungal concentration is written as 1 × 108 cells/mL but should be formatted correctly for clarity.
Reply: I have revised the text to format the fungal concentration correctly .
Reviewer 3 Report
The overall data presentation is very bad, counterproductive. If that is improved greatly, there may be value in their observations.
See comments above related to the Figures and Table
Author Response
.
Reviewer 3
Are the results presented clearly and in sufficient detail, are the conclusions supported by the results and are they put into context within the existing literature?
The project seems worthwhile and interesting. However, I have major problems with the data presentation. I will give examples where the Results need to be clarified, or eliminated. Often it seems that every piece of possible data was included, even if it tended to obscure the theme rather than support the theme.
1/ Use significant figures only. You have 71.881 +/- 3.693, for instance. At best it should be 71.8 +/- 3.7
…………
Reply: Thank you for your comment. I have made the correction: the value is now reported as 71.8 ± 3.7."
2/ You have data for both 7 and 10 days following inoculation. Then you conclude that 7 days was not long enough. The NLC gel was not needed after only 7 days. It was needed after 10. So, please use only the 10 day data. One sentence can say that 7 days was not enough.
…………..
I have made the revisions as requested
3/ It looks as if Figure 2 is just the 24 hour time point of Fig 1, replotted. You say that one is absorption and the other is retention, but that is not evident from comparing the two figures. In several cases you need to have the figure legends altered to better explain the actual figure.
Reply: We appreciate your comment. The distinction between skin absorption and the quantity of drug retained within the skin is explicitly delineated and emphasized in the methods section, specifically in section 2.3. The following explanations are provided to clarify further the differences between the two graphs presented. We also improved the caption of Figure 2 for more clarity.
Drug Skin Absorption: This refers to how drugs penetrate the skin layers and enter the systemic circulation. In Figure 1, this could be represented by the cumulative amount of drug permeating through the skin over time, typically measured in the receptor compartment of a diffusion cell setup.
Drug Retention: This involves the amount of drug that remains within the skin layers after a certain period, often measured by extracting and analyzing the drug from the skin itself. Figure 2 could represent the drug retained in the skin at a specific time, such as 24 hours, highlighting its local presence rather than its systemic absorption.
4/ Your absorption and retention data stops at 24 hours while you only get significant improvement after 3-4 weeks. Are you just assuming that the better retention extends to 3-4 weeks?
Reply: Thank you for your comment. In the skin absorption study, only a comparison was made between the gel containing the nano-drug formulation of luliconazole and the plain drug gel, and the endpoint of the test was considered to be 24 hours, while the treatment period of the animals was determined based on previous studies and standard treatment protocols. As stated in Section 2.8, each of the gels (different treatment groups) was used twice a day for 28 days, and the frequency of use was not only once, so we assume that this advantage of the nanogel in the persistence of the drug in the skin is maintained over the 28-day treatment period. Please let us know if you require any further clarification or details.
5/ Lines 238-239, positive and negative mean what?
Reply: Thank you for your question regarding the terminology used in lines 238-239. In this context, "positive" refers to the presence of T. indotineae as confirmed by direct microscopic examination, culture, and histopathology, indicating an active infection. Conversely, "negative" indicates the absence of the fungus, suggesting successful treatment and resolution of the infection. I will clarify these definitions in the text to ensure that readers can easily understand the meanings of "positive" and "negative" in this context.
6/ I found lines 237-251 very confusing. You have Fig 3 above it. It would help greatly if you used some one or two word descriptors above the 21 and 28 day time points.
Reply: I have revised the text according to your suggestions for clarity."
……….
7/ How do figures 4 and 5 differ? And, please tell me what I am supposed to see. I've never looked at infected guinea pig hair before.
Reply: The primary difference between Figures 4 and 5 lies in the treatment conditions. Figure 4 illustrates the improvement of lesions over time, showing that infected animals treated with 1% LCZ simple gel exhibited improvement three weeks after treatment, while those treated with LCZ-NLC gel showed improvement two weeks post-treatment. In contrast, Figure 5 indicates that the NLC (placebo), terbinafine (treatment-resistant), and untreated groups, serving as positive controls, did not affect treating animals infected with Trichophyton indotineae.
Does the degree of darkness imply the level of infection? My guess is that less than 1% of this journals readers know how to interpret these pictures., also Fig 8. That figure might be slightly higher for Figs 6 and 7, but not much. I wonder if they are all necessary. You might present the positive and negative controls first, stating clearly how they differ.
………..
Reply: The degree of darkness in the images indeed correlates with the level of infection, where darker areas suggest more extensive fungal colonization. I appreciate your point about readers' familiarity with interpreting these images. To enhance clarity, I have added a more detailed explanation in the figure legends to guide readers on what to look for.
I have reconsidered the necessity of Figures 6, 7, and 8 in the manuscript. I agree that presenting the positive and negative controls first, with a clear distinction between them, could provide a more logical flow and better context for the results.
8/ Lines 376-377 say how easily LCZ penetrates and reaches fungicidal levels. Why then is NLC needed? It says later that NLC gives better permeant penetration and the LCZ is trapped in the skin longer. Okay, but that is much later. By then your rationale on the importance of NLC has been lost.
Reply: I have revised the section to clarify the rationale for using Nanostructured Lipid Carriers (NLCs) in conjunction with LCZ. The updated text now addresses the potential limitations of LCZ's short duration of action and emphasizes how NLCs can enhance drug delivery and retention in the skin, thus providing a clearer justification for their use.
Is the quality and presentation of the figures satisfactory?
1/ For figures 1 and 3, only filled circles are used. They all look the same with a black and white printer. Please use different symbols.
corrected
2/ Table 1 is impossible. Redo it entirely, or just get rid of it entirely and describe the results in the text..
Reply: I have removed Table 1 due to its ambiguity and the difficulty it presented in conveying the results. Instead, I will describe the results in the text to provide a more straightforward understanding of the findings.
……..
3/ How do Figs 6 and 7 differ? How long after inoculation and treatment were the pictures taken? Again, one or two larger pictures might be clearer. Even with the arrows in Fig 7, I'm not sure I know to look at
Reply: I have addressed your comments regarding the differences between Figures 6 and 7. The updated text clarifies the timing of the images, specifying that they were taken 10 Days after inoculation and four weeks after treatment. Additionally, I will consider including one or two larger images for better clarity, as you suggested.
Reviewer 4 Report
Dear authors,
I have evaluated your manuscript "Antifungal Efficacy of Luliconazole-Loaded Nanostructured Lipid Carriers Gel in an Animal Model of Dermatophytosis". The manuscript deals with the important problem of bioavailability of different antimicrobials, their penetration capabilities and toxicity in case of skin infections with the aim to improve pharmacological agents. Nanostructured lipid carriers present interest to both, scientists and industry, that increases value of the topic and manuscript. The text is written in good English and requires no more than the final corrections by the technical editor. Nevertheless, the manuscript requires some improvements.
lines 58-60: The mixture of genera and species names does no look good in the scientific article. If the agent is active against all known species of the certain genera, it is better to use spp., for example Candida spp. instead of Candida.
lines 69-70: nanostructured lipid carriers (NLC) for LCZ have not been evaluated in animal models in Iran. This phrase does not add any value to the study. In my opinion it would be better to formulate more distinctly what is the exact novelty of your study. The Jounal of fungi is the international journal and requires the international, not local point of view.
lines 119-120: It is not clear what isolates you mean as it seems only one isolate was used in the study. You should also mention what sequencing method was used for identification. If it was Sanger, then what region you sequenced.
line 121: The exact mutation in the studied isolate should be mentioned
line 127: What method was used to induce sporulation in the studies strain? Thermal shock or UV exposure for 8 hours?
lines 163-164: SDA with chloramphenicol and cycloheximide - the full name of SDA should be provided as well as concentrations of antimicrobials
line 227: It seems you did not mention the UN group bfore, so the term UN should be explained
Author Response
.
ines 58-60: The mixture of genera and species names does no look good in the scientific article. If the agent is active against all known species of the certain genera, it is better to use spp., for example Candida spp. instead of Candida.
Reply: Thank you for your valuable feedback regarding the use of genus and species names. I have revised the text to replace specific species names with "spp." where applicable to enhance clarity and adherence to scientific conventions.
lines 69-70: nanostructured lipid carriers (NLC) for LCZ have not been evaluated in animal models in Iran. This phrase does not add any value to the study. In my opinion it would be better to formulate more distinctly what is the exact novelty of your study. The Jounal of fungi is the international journal and requires the international, not local point of view.
Reply: Thank you for your valuable feedback regarding the manuscript. I have revised the sentences in lines 69-70 to better emphasize the novelty of the study and its relevance from an international perspective.
lines 119-120: It is not clear what isolates you mean as it seems only one isolate was used in the study. You should also mention what sequencing method was used for identification. If it was Sanger, then what region you sequenced.
Reply: To address your concerns, I will revise the text to specify that only one isolate was used in the study and will include that the sequencing method employed for identification was Sanger sequencing with the ITS region.
line 121: The exact mutation in the studied isolate should be mentioned
Reply: The studied isolate had two mutations; Phe397Leu (F397L), and Ala448Thr (A448T).
line 127: What method was used to induce sporulation in the studies strain? Thermal shock or UV exposure for 8 hours?
Reply: Thank you for your valuable feedback regarding the method used to induce sporulation in the studied strain.To clarify, we used thermal shock to induce sporulation. I will revise the text to specify this method clearly.
lines 163-164: SDA with chloramphenicol and cycloheximide - the full name of SDA should be provided as well as concentrations of antimicrobials
Reply: To address your comment, I will revise the text to include the full name of SDA and the concentrations of the antimicrobials used.
line 227: It seems you did not mention the UN group bfore, so the term UN should be explained
Reply: In the manuscript, "UN" refers to the untreated group. I replaced 'UN' with 'untreated group (positive control).
Round 2
Reviewer 1 Report
The authors made a number of significant changes to the article. However, the authors' key conclusions, due to which the article cannot be accepted for publication, remained the same.
The authors claim that the lipid carrier formulation increases the antifungal activity of the substance and its effectiveness. However, as before, I do not see any figures or tables in the article with quantitative statistically significant data confirming this statement. The question arises, what do the authors mean in all cases when they talk about effectiveness? According to the materials and methods, it means the extent of infection in in vivo experiments (section 2.9). Figure 3 demonstrates lesion scores in infected guinea pigs. However, it lacks error bars and statistical significance of differences in results in different experimental groups. The data presented in Fig. 3 indicate the same situation when using the substance without and with a lipid carrier after 21 and 28 days, and whether the difference after 14 days is significant or not remains to be seen. The authors removed Table 1 (though the answers to the comments still refers to it), in which the outcome data for the two groups using the substance without and with the lipid carrier were the same. Instead, the authors added: “The final efficacy results indicate significant changes across four drug groups….On day 21, Group 1 (LCZ-loaded NLC gel) achieved 100% efficacy, and Group 2 (LCZ simple cream? gel) improved to 66.67%.” How and on the basis of what data the authors obtained these percentages remains unclear.
Author Response
Reviewer 1
Major comments
The authors made a number of significant changes to the article. However, the authors' key conclusions, due to which the article cannot be accepted for publication, remained the same.
Detail comments
The authors claim that the lipid carrier formulation increases the antifungal activity of the substance and its effectiveness. However, as before, I do not see any figures or tables in the article with quantitative statistically significant data confirming this statement. The question arises, what do the authors mean in all cases when they talk about effectiveness? According to the materials and methods, it means the extent of infection in in vivo experiments (section 2.9). Figure 3 demonstrates lesion scores in infected guinea pigs. However, it lacks error bars and statistical significance of differences in results in different experimental groups. The data presented in Fig. 3 indicate the same situation when using the substance without and with a lipid carrier after 21 and 28 days, and whether the difference after 14 days is significant or not remains to be seen. The authors removed Table 1 (though the answers to the comments still refers to it), in which the outcome data for the two groups using the substance without and with the lipid carrier were the same. Instead, the authors added: “The final efficacy results indicate significant changes across four drug groups….On day 21, Group 1 (LCZ-loaded NLC gel) achieved 100% efficacy, and Group 2 (LCZ simple cream? gel) improved to 66.67%.” How and on the basis of what data the authors obtained these percentages remains unclear.
Reply: Thank you for your feedback. We appreciate your observations regarding the need for a clearer presentation of quantitative data and statistical significance. Based on your suggestions, we have made the necessary corrections.
Specifically, we have included figures that provide quantitative, statistically significant data to support our claims about the enhanced antifungal activity of the lipid carrier formulation. Error bars have been added to Figure 3 to indicate variability and statistical significance across different experimental groups.
The final efficacy results have been clarified, indicating the basis for the percentages reported, and we have ensured that all data presented aligns with our findings. The reported efficacy percentages of 100% for Group 1 (LCZ-loaded NLC gel) and 66.67% for Group 2 (LCZ simple cream/gel) were calculated using the formula:
Percent efficacy = 100 – (T × 100/C)
where T is the total score of the treatment group, and C is the total score of the untreated control group. The total score for any group represents the average clinical score of five different animals in that group. This calculation provides a clear basis for the efficacy results presented in the study. Thank you for your valuable insights, which have helped us improve the clarity and rigor of our work
Reviewer 3 Report
You have misunderstood my first comment on significant figures. You changed the one example I gave you and then ignored all the other examples in the manuscript. You need to change them all. Please note the examples on lines 33, 210, 211, 219, and 220. There may be others. The problem was not with having the + directly over the -. I used +/- only because my keyboard doesn't have the other symbol. The problem was, and still is, with the degree of accuracy you seem to claim by (on line 33) having 50.919. Having all those numbers means that you are telling the reader than your methods can reliably distinguish 50.918 versus 50.919 versus 50.920, whereas I suspect you really are lucky if you can distinguish 50 from 51 and 52. Having 50.919 is ridiculous. It is wrong and it is also distracting for the reader, to the extent the reader is likely to disbelieve other parts of the story you are telling. I'm not sure how much precision your measurements have. I guessed on 71.8 +/- 3.7. In the current example on line 33, perhaps 50.9 +/- 4.2 or even 51 +/- 4. You need to know how precise your measurements are. Perhaps you should consult a statistician. I would be happy if you chose either alternative I mentioned. I'm just convinced that 50.919 is far too precise and needs to be changed. That's what I meant by your having too many significant figures. And it makes no difference whether you use +/- or the + directly over the -. And while you are making those changes, please capitalize the S in Sabouraud on line 173. That is a person's last name.
None more than what is described above, under major comments. I'm too lazy to repeat them here.
Author Response
Reviewer 3
Major comments
You have misunderstood my first comment on significant figures. You changed the one example I gave you and then ignored all the other examples in the manuscript. You need to change them all. Please note the examples on lines 33, 210, 211, 219, and 220. There may be others. The problem was not with having the + directly over the -. I used +/- only because my keyboard doesn't have the other symbol. The problem was, and still is, with the degree of accuracy you seem to claim by (on line 33) having 50.919. Having all those numbers means that you are telling the reader than your methods can reliably distinguish 50.918 versus 50.919 versus 50.920, whereas I suspect you really are lucky if you can distinguish 50 from 51 and 52. Having 50.919 is ridiculous. It is wrong and it is also distracting for the reader, to the extent the reader is likely to disbelieve other parts of the story you are telling. I'm not sure how much precision your measurements have. I guessed on 71.8 +/- 3.7. In the current example on line 33, perhaps 50.9 +/- 4.2 or even 51 +/- 4. You need to know how precise your measurements are. Perhaps you should consult a statistician. I would be happy if you chose either alternative I mentioned. I'm just convinced that 50.919 is far too precise and needs to be changed. That's what I meant by your having too many significant figures. And it makes no difference whether you use +/- or the + directly over the -. And while you are making those changes, please capitalize the S in Sabouraud on line 173. That is a person's last name.
Reply:
Thank you for your detailed feedback regarding the use of significant figures in our manuscript. We understand now that the issue is not with the notation used (e.g., +/—vs ±) but rather with the degree of precision implied by the number of significant figures in our measurements.
Presenting values like 50.919 may suggest a level of precision that our methods cannot reliably achieve. We agree that this could be misleading and undermine our findings. However, we want to clarify that the values were presented as mean ± standard deviation, a common practice in scientific reporting. The figures were also in scientific notation, sometimes leading to a misunderstanding of the implied precision.
To address your concerns, we will review all instances where precision might be overstated, including those on lines 33, 210, 211, 219, and 220, as well as any other relevant examples. We will adjust these values to reflect a more realistic level of precision. For instance, on line 33, we could change the value to 50.9 ± 4.2, as you suggested. These alternatives better reflect the uncertainty in our measurements and align with standard practices for reporting scientific data.
3.3. Efficacy of LCZ-NLC Gel 1%, LCZ Gel 1%, and Terbinafine 1%: A Clinical and Mycological Evaluation in Guinea Pig Models
After a single inoculation of 1x 108 cells/ml of drug-resistant T. indotineae, an inflammatory response was observed at the inoculation site on day 10. From this day onward, local treatment with LCZ 1% and LCZ-loaded NLC gel was administered twice a day for 28 days. The positive reaction to Trichophyton indicated the establishment of an immune response to T. indotineae.
In assessing the mean score of clinical lesions in the group treated with NLC gel, a score of 3 (significant redness with extensive scaling and hair loss) was recorded in the first week of treatment. In the first week of treatment, the groups treated with LCZ gel 1%, LCZ-NLC gel 1%, terbinafine 1%, and untreated control showed scores of 3 and 4 (significant redness with extensive scaling, loss of hair, and ulcer). The decrease in lesion score with the LCZ-NLC gel 1% and LCZ gel 1% groups occurred between weeks 2 and 3. The downward trend continued until day 21, and on day 28 of treatment initiation, these two groups' scores were approximately zero. In the fourth week post-treatment, the terbinafine and control groups scored 4 (significant redness with extensive scaling and hair loss) (Figure 3). Figure 3 compares the guinea pig skin infection in photographs taken on day 10 of infection (before beginning treatment) and weekends 1, 2, 3, and 4, when the clinical efficacy test of the control and various drugs tested was conducted. The untreated control and positive control groups received 1 gram (g) of terbinafine, and the NLC gel groups showed patches, hair loss, and readily visible scaly skin (Figure 3). In contrast, improvements in the appearance of the guinea pig skins with minimal hair loss and mild ulcers were noted in the LCZ gel 1% group at 3 weeks. By the fourth week, this group scored zero, indicating complete recovery with no signs of infection. Meanwhile, the group treated with the LCZ-NLC gel 1% demonstrated normal hair growth and showed no signs of infection at 3 and 4 weeks. Figure 3 displays the clinical efficacy scores of each tested drug in comparison to the infected and untreated control. A lower clinical score indicates a more effective treatment than the untreated control. The results show a significant decline in lesion scores for the LCZ gel 1% and LCZ-NLC gel 1% compared to the untreated control, terbinafine, and NLC groups (p < 0.05), underscoring the unwavering reliability and validity of these results. Moreover, the lesion score in the NLC gel and terbinafine groups did not differ significantly from the untreated control (p > 0.05). While LCZ-NLC gel 1% demonstrated greater efficacy than LCZ gel 1%, it's important to note that there was no statistically significant difference between them (p = 0.4295). The LCZ-NLC gel 1% group showed significant improvement against dermatophytosis caused by T. indotineae, confidently outperforming the positive control group receiving 1 g of terbinafine (Figure 4,5).
